# Compression of Multibeam Echosounders Bathymetry and Water Column Data

Aniol Martí [1,*], Jordi Portell [1,2], David Amblas [3], Ferran de Cabrera [4], Marc Vilà [4], Jaume Riba [4] and Garrett Mitchell [5,6]

1   DAPCOM Data Services, Vilabella Centre de Negocis, Vilabella 5-7, E-08500 Vic, Spain; jportell@fqa.ub.edu
2   Institut de Ciències del Cosmos (ICCUB), Universitat de Barcelona (IEEC-UB), Martí i Franquès 1, E-08028 Barcelona, Spain
3   GRC Geociències Marines, Departament de Dinàmica de la Terra i de l'Oceà, Universitat de Barcelona (UB), E-08028 Barcelona, Spain; damblas@ub.edu
4   Signal Processing and Communications Group (SPCOM), Departament de Teoria del Senyal i Comunicacions, Universitat Politècnica de Catalunya (UPC), E-08034 Barcelona, Spain; ferran.de.cabrera@upc.edu (F.d.C.); marc.vila.insa@upc.edu (M.V.); jaume.riba@upc.edu (J.R.)
5   Fugro USA Marine Inc., 6100 Hillcroft Avenue, Houston, TX 77081, USA; gmitchell@fugro.com
6   Center for Coastal and Ocean Mapping, University of New Hampshire, Durham, NH 03824, USA
*   Correspondence: aniol.marti@dapcom.es

**Abstract:** Over the past decade, Multibeam Echosounders (MBES) have become one of the most used techniques in sea exploration. Modern MBES are capable of acquiring both bathymetric information on the seafloor and the reflectivity of the seafloor and water column. Water column imaging MBES surveys acquire significant amounts of data with rates that can exceed several GB/h depending on the ping rate. These large file sizes obtained from recording the full water column backscatter make remote transmission difficult if not prohibitive with current technology and bandwidth limitations. In this paper, we propose an algorithm to decorrelate water column and bathymetry data, focusing on the KMALL format released by Kongsberg Maritime in 2019. The pre-processing stage is integrated into FAPEC, a data compressor originally designed for space missions. Here, we test the algorithm with three different datasets: two of them provided by Kongsberg Maritime and one dataset from the Gulf of Mexico provided by Fugro USA Marine. We show that FAPEC achieves good compression ratios at high speeds using the pre-processing stage proposed in this paper. We also show the advantages of FAPEC over other lossless compressors as well as the quality of the reconstructed water column image after lossy compression at different levels. Lastly, we test the performance of the pre-processing stage, without the constraint of an entropy encoder, by means of the histograms of the original samples and the prediction errors.

**Keywords:** Multibeam Echosounders (MBES); lossless data compression; lossy data compression; water column; sea imaging

## 1. Introduction

Advances in sonar and computing technologies have led to major advances in ocean exploration over the last decades. In addition to bathymetric and seafloor reflectivity data, Multibeam Echosounders (MBES) are now capable of imaging the water column. This new mapping capability has introduced new techniques for marine research, including the direct mapping of fish shoals, gas seeps or other mid-water column targets [1,2]. Some MBES even support multi-frequency data acquisition in combination with water column data, hence increasing their detection capability [3,4]. Concurrently, ping rates and the number of beams in new multibeam sonars continuously increase. All these improvements in MBES technology imply high data acquisition rates and encompass the massive collection of data, which can easily be one order of magnitude larger than those in standard bottom detection

bathymetric surveys. This poses technical and logistical challenges to MBES users that, for this reason, often opt to systematically turn off the recording of water column data [5]. Moreover, this fact limits the autonomy of unmanned surface and underwater vehicles due to data storage needs and battery drain, and it dramatically increases the cost of remote assistance relying on expensive satellite communications.

A potential solution to this challenging question in ocean research is data compression. For this reason, there are some studies addressing this problem [6–9]. However, only a few propose lossless algorithms [10–12]. In Reference [11], the authors focus on the MBES data from the manufacturer RESON, which stores samples in linear units. As part of their algorithm, Moszynski et al. [11] propose converting them to decibels with a 0.5 dB precision, which is higher than the precision of the echosounder, hence guaranteeing a lossless compression. In the following sections, we will see that this step is not always needed, as some manufacturers, such as Kongsberg Maritime, already store samples logarithmically and with a resolution of 0.5 dB.

In Reference [12], a tailored and optimized solution to compress MBES data based on stream partitioning was presented. This consisted of adding a new stage to the Fully Adaptive Prediction Error Coder (FAPEC), a compressor for space applications. In their case, Portell et al. [12] focused on the Water Column Data (WCD) from Kongsberg Maritime. However, in 2019, with the introduction of the latest generation of EM-series multibeam sonars, this manufacturer released a new data format called KMALL. The format contains high-resolution data, is structured in datagrams containing different kinds of data, and is generic enough to facilitate future updates. Due to its novelty, no tailored compression algorithms have been developed yet.

In this paper, we focus on developing an algorithm to compress the water column data from the KMALL format. In addition, we also perform an analysis of several `.kmall` files and propose a new algorithm for the datagrams that contain bathymetry data.

This paper is structured as follows. Section 2 explains the KMALL data format. As will be shown hereafter, KMALL compression has been tackled in a relatively generic manner that could be applied to other formats or vendors as well. Section 3 provides a brief description of the FAPEC data compressor. Section 4.1 describes the data analysis performed before designing the algorithm. Sections 4.2–4.6 present the implementation of the bespoke pre-processing stage. Section 5 describes the test files and software used. Section 6 shows the results of the proposed algorithm. Finally, Section 7 presents our conclusions and states future work lines.

## 2. The KMALL Data Format

Before describing the structure of a KMALL file, we shall provide five definitions related to the echosounder's performance. We define a *ping* as the set of acoustic pulses transmitted at approximately the same time. Each of these pings is formed by pulses directed into different angles, which are called *beams*. The number of beams per ping depends on the sonar model. The number of samples per beam depends on the angle and the depth. Each transmit pulse defines an across sector called a *transmit sector*. Simultaneously, a *receiver fan* is defined as the set of received soundings from one unit. Then, based on the transmit sectors and the receiver fans, a *swath* is defined as the complete set of across-track data.

The KMALL format is the successor of the Kongsberg Maritime `.all` format. KMALL is a generic format with high-resolution data and is also structured in datagrams. This structure is designed to avoid breaking existing decoders when updating the format. Datagrams are usually stored in binary little-endian format with a 4-byte alignment [13]. Each datagram starts with a generic header containing the datagram size in bytes and a 4-byte identifier. KMALL datagrams do not have any size constraints, as opposed to the `.all` format, which is limited to 64 kB due to the maximum size of UDP packets. The main advantage of this feature is that pings are not split between datagrams.

The size of a KMALL file is not limited either. The echosounder operator decides when the file being recorded should end. Each file contains several datagrams, the most relevant ones being Multibeam Water Column (MWC), Multibeam Raw Range and Depth (MRZ) and Sensor Attitude Data (SKM) datagrams. This mixture of data types in the same file makes it difficult for general-purpose data compressors such as *gzip* to identify data statistics. For this reason, they do not perform as well as one could expect.

In this study, we analyze multibeam data acquired with Kongsberg EM2040, EM712 and EM304 MBES models, kindly provided by Kongsberg Maritime and Fugro. EM2040 is a high-resolution shallow water multibeam system operating at sonar frequencies in the 200–400 kHz range with an angular coverage up to 200° and pulse lengths as short as 14 µs. EM712 can survey depths up to 3500 m and operates at frequencies in the 40 to 100 kHz range with a swath coverage sector up to 140° and pulse lengths up to 0.2 ms, while EM304 allows mapping waters as deep as 8000 m with an operating frequency between 26 and 34 kHz and a pulse length of 0.4 to 200 ms.

### 2.1. MWC Datagrams

Multibeam Water Column (MWC) datagrams contain information about the water column. They are usually stored in `.kmwcd` files, although they can also be stored in `.kmall` files together with the rest of the datagrams (remarkably, MRZ). Each MWC datagram is composed of several elements or data structures, as can be seen in Figure 1.

| Header 20 [B] | Partition 4 [B] | CmnPart 12 [B] | TxInfo 12 [B] | SectorData $9 \cdot 16$ [B] | RxInfo 16 [B] | BeamData $N_B \cdot (16 + N_S \cdot (1 + \text{phaseMode}))$ [B] |
|---|---|---|---|---|---|---|

**Figure 1.** MWC datagram as described by Kongsberg (410224 Revision H). Sizes of data elements not to scale.

We now describe the information contained in each element. The *Header* contains information such as the MWC datagram size in bytes, the type of datagram (#MWC), the echosounder ID and a time stamp. The *Partition* element has two fields: number of datagrams and datagram number. They are used if some datagram is bigger than 64 kB and it needs to be split for the transmission. Note that a `.kmall` file will always have both fields equal to one. In the *CmnPart*, there is information about the transmitter and the receiver such as the number of pings and the swaths per ping. *TxInfo* contains the number of transmit sectors and the size in bytes of each sector. In *SectorData*, there are as many elements as transmit sectors. Each element contains information about the angle and frequency of each sector. In *RxInfo*, we find information about the beams such as the number of beams and a flag telling if there are high-resolution beam phase data, low-resolution beam phase data or only beam amplitude samples. Finally, *BeamData* contains the number of samples in the beam and a pointer to the beam samples. Amplitude samples (see Figure 2) are integer values, coded as signed 8-bit samples in our case and with a 0.5 dB precision. Phase samples may be stored as 8-bit integers (low resolution) or 16-bit integers (high resolution). Depending on this, *phaseMode* can be zero (no phase data), one (8-bit) or two (16-bit).

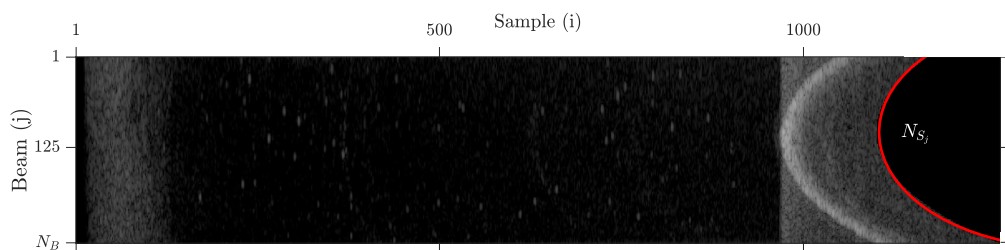

**Figure 2.** Raw representation as a 1270 × 256-pixel picture of water column amplitude samples from a ping acquired by a Kongsberg EM2040 echosounder. Note that the black area on the right does not contain any sample.

After describing the structure of MWC datagrams, it is clear that $N_B$ from Figure 1 is the number of beams contained in RxInfo, and $N_S$ is the number of samples contained in BeamData. We remark that the number of samples per beam changes along the ping, which means that typical image compression algorithms cannot be easily applied to this kind of data. In Figures 2 and 3, we illustrate, respectively, a raw representation of water column amplitude samples and the variability of the number of samples per ping.

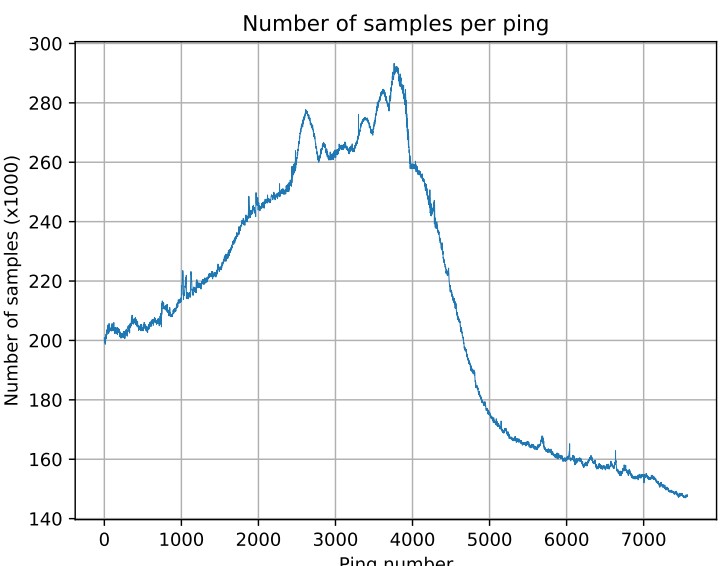

**Figure 3.** Number of samples per ping in a test file from a Kongsberg EM2040 echosounder.

### 2.2. MRZ Datagrams

Multibeam Raw Range and Depth (MRZ) is the most common type of multibeam datagram from Kongsberg Maritime. It replaces several old datagrams from the `.all` format: raw range (N 78), depth (XYZ 88), seabed image (Y 89) datagram, quality factor (O 79) and runtime (R 52).

MRZ datagrams are stored in `.kmall` files. Usually, a file with MRZ datagrams does not contain MWC datagrams, although some exceptions may occur. MRZ datagrams have a maximum size of 366.456 kB, yet they can be smaller. Its structure is shown in Figure 4.

| Header 20 [B] | Partition 4 [B] | CmnPart 12 [B] | PingInfo 152 [B] | SectorInfo 9 · 48 [B] | RxInfo 32 [B] | ExtraDetClassInfo 11 · 4 [B] | Sounding 2048 · 120 [B] | SeabedImage 60,000 · 2 [B] |
|---|---|---|---|---|---|---|---|---|

**Figure 4.** MRZ datagram as described by Kongsberg (410224 Revision H). Sizes of data elements not to scale.

The first three data structures of an MRZ datagram are the same as in an MWC datagram. *PingInfo* cotains common information for all the beams in the current ping. It also contains the number of SectorInfo elements and its size in bytes. The number of elements will be smaller than nine and the size will be smaller than 48 bytes. Each *SectorInfo* element contains information about the transmit sectors, such as the frequency or the angle. *RxInfo*, as in the MWC case, contains information from the receiver. However, the structure type is not the same. It also contains the number of bottom soundings, the number of extra detections (soundings in the water column), the number of extra detections classes and the bytes in each class. In *ExtraDetClassInfo*, we find the number of detections of each class. Note that the sum of all these fields must equal the number of extra detections given in RxInfo. The *Sounding* structure contains data for each sounding such as reflectivity and two-way travel time. It also contains the number of seabed imaging samples, *SInumSamples*, in the last struct.

### 2.3. SKM Datagrams

Sensor Attitude Data (SKM) is another common type of datagram from the KMALL format. It may contain several sensor measurements, and it is usually present in `.kmall` and `.kmwcd` files. Its structure is much simpler than that of MRZ or MWC, as can be observed in Figure 5.

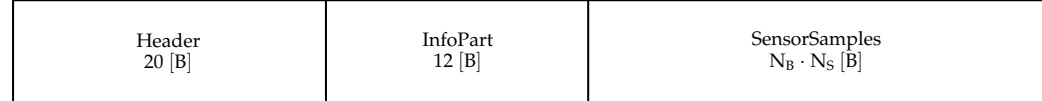

**Figure 5.** SKM datagram as described by Kongsberg (410224 Revision H). Sizes of data elements not to scale.

As usual, the first structure is a header to identify the datagram type and size. The second one contains information about the *SensorSamples* such as the number of samples $N_S$ or the number of bytes per sample $N_B$. Finally, the last structure contains samples retrieved by the sensors such as attitude data. Most of the fields are floating point numbers; hence, we need to pay special attention here when performing the pre-processing.

### 3. The FAPEC Data Compressor

Fully Adaptive Prediction Error Coder (FAPEC) is a versatile and efficient data compressor originally designed for space missions [14]. The main advantages of FAPEC are its high computing performance and its compression resilience in front of noise or data outliers.

FAPEC is based on a two-stage approach: a pre-processing stage followed by the entropy coder. In fact, its name comes from this architecture as, usually, the first stage is some type of predictor that tries to estimate the true samples and generates a prediction error. Then, the error is sent to the entropy coder instead of the original samples (thereof the Prediction Error Coder naming).

Formally, given an input sample $x_i$ and an estimator $\hat{x}_i$, the value sent to the entropy coder is:

$$e_i = x_i - \hat{x}_i. \tag{1}$$

Restoring the original value in decompression is straightforward:

$$x_i = \hat{x}_i + e_i. \tag{2}$$

Note that in a few stages, some flags are also sent to the entropy coder. Figure 6 illustrates the general approach, where $T()$ is a generic transformation.

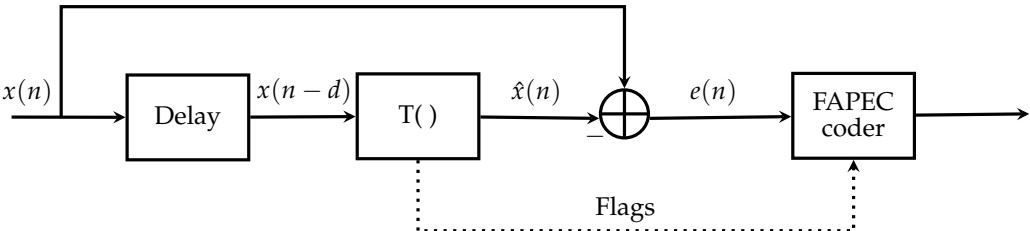

**Figure 6.** The FAPEC data compressor structure.

It is well known [15] that the Shannon entropy fulfills the following bound:

$$H(\mathbf{X}) \leq \sum_n H(x[n]) \tag{3}$$

where $\mathbf{X}$ is a block of data and $x[n]$ are the samples of the block.

Equation (3) suggests that the optimum approach is to design an entropy coder that operates on data blocks. However, the design of such encoder is too complex. For this reason, pre-processing stages are used. Their main purpose is to reduce the memory of the source $\mathbf{X}$, that is, whitening the error sequence $e(n)$ to allow the entropy coder to work sample by sample. If the pre-processing stage is linear, as in this case, it is only able to decorrelate the input data but not to remove the dependence. Nevertheless, if the prediction error is Gaussian, it is known that uncorrelatedness and independence are equivalent; hence, decorrelation is enough. In other words, the pre-processing stage performance is critical to the overall compressor efficiency.

FAPEC features several pre-processing algorithms such as a basic differential, multi-band prediction [16] or even a wavelet transform. In FAPEC 19.0 a pre-processing stage for `.wcd` files, the former water column format by Kongsberg Maritime, was included. In the latest version, FAPEC 22.0, the new KMALL stage here described is included, as well as other new stages for multispectral images or audio and IQ data.

In order to achieve a good adaptation in front of outliers, FAPEC works on data blocks of, typically, 128 samples. The core analyzes every block of prediction errors and determines the optimum coding tables. Finally, it calls the coding kernel, which generates a variable-length code for every given prediction error.

As part of this coding kernel, FAPEC features a run-length encoding algorithm that identifies a series of null prediction errors (that is, perfect predictions) and codes them efficiently. This is a useful feature that should be taken into account when designing a pre-processing stage. Remarkably, it allows for significantly better ratios when using the lossy option (see Section 4.5), given the quantized nature of the resulting prediction errors.

Finally, for robustness and computing performance purposes, FAPEC compresses data in chunks (usually between 64 kB and 8 MB). That is, input data are split into several chunks of the same size, and each chunk is processed independently of the others. Note that data chunks are not the data blocks mentioned earlier, as the former are treated completely independently both in the pre-processing and the coder, and the latter are the length of the adaptive block in the coder.

## 4. FAPEC Tailoring for the KMALL Data Format

### 4.1. Preliminary Data Analysis

In order to design a new pre-processing stage, we need to know the structure of our data. In this case, we have decided to narrow it down to the datagram level. That is, we perform a preliminary analysis to find the most relevant datagrams in a `.kmall` (or `.kmwcd`) file. We have proceeded as follows:

1. Implement a basic decoder of the KMALL format in Python.
2. Implement a Python class that acts as top level API for the KMALL files.
3. Write a Python script that takes a directory and an extension and returns the percentages of every datagram type on each file.

The distribution of the datagrams strongly depends on the captured scene. That is, on the water column and seafloor characteristics, number of beams, resolution, ping rate and other physical properties of the echosounder. In our dataset, the result is that in .kmall files, MRZ datagrams percentage oscillates between 70% and 95% of the total file size. The remaining space is mainly occupied by SKM datagrams. In .kmwcd files MWC, datagrams are more than 99% of the file size. Finally, in files containing MRZ and MWC datagrams, these occupy between 89% and 92% of the total size, with the remaining being mainly SKM datagrams. More detailed information about the used dataset is available in Section 5.

Clearly, the most interesting datagram types are MRZ and MWC, although in some situations SKM, datagrams must also be taken into account. Hence, we propose three different algorithms, one for each datagram type.

Even though this paper is mainly focused on the KMALL data format, the algorithms presented in the following sections can be applied to similar data structures from other formats or vendors. Before continuing, we must pay attention to the FAPEC data chunking and the datagram identification. The former is a useful feature, although it might be risky because it may result in splitting a datagram into different chunks, hence making it difficult to read. For this reason, we will automatically reduce the actual chunk size (with respect to the value given by the user) to ensure an integer number of datagrams per chunk. With this approach, the observed efficiency in the identification and handling of complete datagrams in a chunk is between 74% and 100%, with an average of 95%. With this constraint, in order to identify a datagram type, we just need to read the corresponding field in the header. Then, if it is an MWC, an MRZ or an SKM datagram, we will proceed with one of the following algorithms. Otherwise, samples are directly sent to the entropy coder.

*4.2. MWC Pre-Processing Stage*

In this section, we propose an algorithm to decorrelate MWC datagrams. We only focus on the BeamData subdatagram, as the others are negligible in comparison.

If we represent the amplitude samples as a raw rectangular image (see Figure 2), we can see that correlation between beams is higher than between samples (i.e., the image is smoother from top to bottom than left to right). Although this is purely qualitative, it matches with the algorithm designed in [12], and it can be proved to be true by applying a simple differential first by rows and then by columns and finally comparing the results. In the picture, we can also see that the number of samples per beam is not constant, but it has a parabolic behavior.

In the implementation for .wcd files in [12], given a sample $S_{i,j}$ in the position $i$ of the beam $j$, the vertex coordinates $(V_x, V_y)$ and the number of beams $N_B$, prediction errors $E_{i,j}$ in the rectangular region are calculated:

$$E_{i,j} = S_{i,j} - S_{i,j-1}, \quad 0 \le i < V_x, \quad 1 \le j < N_B. \tag{4}$$

In other words, the samples from the rectangular region are predicted from the samples in the same position from the previous beam.

The remaining samples are predicted following a simple differential between them and a correction coefficient. Formally, given $N_{S_j}$ the number of samples in the beam $j$, then

$$E_{i,j} = S_{i,j} - S_{i-1,j} - C_{i,j}, \quad Vx \le i < N_{S_j}, \quad 1 \le j < N_B \tag{5}$$

where $C_{i,j}$ is given by:

$$C_{i,j} = \frac{1}{16}(7E_{i-1,j} + 6E_{i-2,j} + 4E_{i-3,j} + 3E_{i-4,j}). \tag{6}$$

As the reader may have noticed, these equations are not defined for $j = 0$ (i.e., the first beam). The reason is that the system must be causal; therefore, the first beam is predicted from the previous sample in the same beam:

$$E_{i,0} = S_{i,0} - S_{i-1,0}, \quad 1 \le i < N_{S_0}. \tag{7}$$

The algorithm we propose in this section consists of an adaptation from that of [12] but with some improvements. We also follow the approach of predicting samples from the previous beam, but we do not restrict it to the rectangular region. Instead, we do that for all the samples that have a sample above. Formally,

$$E_{i,j} = S_{i,j} - S_{i,j-1}, \quad 0 \le i < \min(N_{S_{j-1}}, N_{S_j}), \quad 1 \le j < N_B. \tag{8}$$

The remaining samples are predicted from the previous samples. That is,

$$E_{i,j} = S_{i,j} - S_{i-1,j}, \quad \min(N_{S_{j-1}}, N_{S_j}) \le i < N_{S_j}, \quad 1 \le j < N_B. \tag{9}$$

The prediction errors for the first beam are also calculated with Equation (7), with the addition that

$$E_{0,0} = S_{0,0} + 128. \tag{10}$$

Now we are interested in observing if this algorithm also works for the phase information present in MWC datagrams. Notice that the former stage for `.wcd` was not designed to work with phase samples. In Figure 7, we show the phase corresponding to the amplitudes in Figure 2. As the reader can see, this information is quite stochastic and much more difficult to predict than amplitude samples. Our final decision has been to apply a simple differential between samples (i.e., Equation (9)).

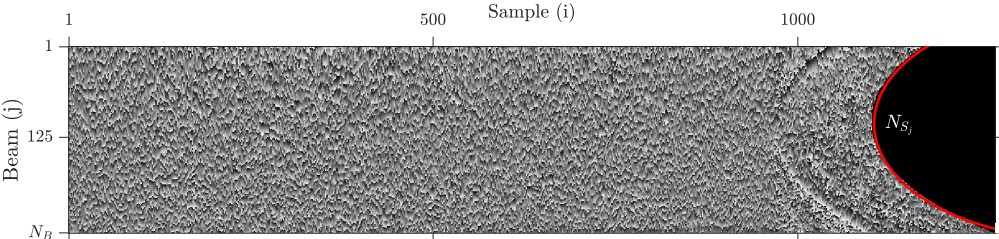

**Figure 7.** Raw representation of MWC phase data from an EM2040 echosounder.

We should remark that well-known algorithms such as nearest-neighbor or a generic linear predictor of order $Q$ have been tested, but they are slower and also have a worse compression ratio. Our approach is simple (thus fast) and already provides good ratios, also allowing us to easily introduce losses (see Section 4.5).

### 4.3. MRZ Pre-Processing Stage

The second algorithm we propose is focused on MRZ datagrams, the most used ones in oceanographic research and industry, yet no tailored compression algorithms have been developed so far.

Following the same procedure as in the previous section, the first step is to identify the larger subdatagrams. Assuming a maximum size datagram of 366.456 kB, sounding subdatagrams can take up to 245.760 kB, and seabed imaging up to 120 kB (see Figure 4). That is, in maximum size MRZ datagrams, sounding and seabed imaging subdatagrams occupy 99.8% of the total size. Hence, we only focus on these two subdatagrams while the others are directly sent to the entropy coder, as we did in MWC compression. Note that in the worst case (just one sounding and no seabed imaging) it still represents 74% of the total size.

The algorithm we propose exploits the periodicity of the sounding subdatagrams. Each sounding subdatagram contains several numerical fields (different types of integers and floats, see [13]), but the same field is located every 120 bytes, as these subdatagrams have a fixed length. For simplicity, efficiency and speed, we process each subdatagram as 16-bit samples with an interleaving step of $I = 60$. Then, our approach consists in computing the prediction errors by subtracting the previous subdatagram from the current one. Furthermore, we reorder the prediction errors in such a way that the same fields from all subdatagrams are placed together. Reordering ensures that long sequences of the same data type are sent to the entropy coder, which improves the performance. Formally, given $N_{Sd}$ soundings with $I$ 16-bit samples $S_{i,j}$,

$$E_{i+j \cdot N_{Sd}} = S_{i,j} - S_{i-1,j}, \quad 1 \leq i < N_{Sd}, \quad 0 \leq j < I. \tag{11}$$

Note that the first sounding $i = 0$ is directly copied to the prediction errors sequence:

$$E_{j \cdot N_{Sd}} = S_{0,j}, \quad 0 \leq j < I. \tag{12}$$

Before continuing, we briefly explain why using 16-bit samples and reordering the prediction errors improves the performance of the entropy coder. FAPEC supports sample sizes from 4 to 28 bits. In order to keep the synchronism, we must choose a value that is a multiple of 8 bits. Moreover, due to the presence of float numbers, the chosen sample size must be a divisor of 32. Hence, the only possible choices are 8 bits or 16 bits. Finally, 16 bits are preferred because, for a given sample rate, these maximize the bit rate at the compressor output. Note that using 16 bits means splitting floats into two parts: the first one (sign, exponent and 7 bits of mantissa) not changing much between soundings [17] and the second one being mostly noise as the resolution of an echosounder is below that of floating point numbers. With this approach, FAPEC will be able to efficiently compress half of the floats (and most of the integer values), whereas the other half will be barely compressed.

The importance of reordering samples is related to the data blocks used by the FAPEC core (see Section 3 or [14]). The order inside a block is relevant for run-length encoding. Furthermore, if similar samples fall in different blocks, then the block entropy will be higher, and hence, the performance will be lower.

Finally, seabed imaging samples are predicted from the previous sample:

$$E_i = S_i - S_{i-1}, \quad 0 \leq i < N_{Sb} \tag{13}$$

where $N_{Sb}$ is obtained by summing all the *SInumSamples* fields in the sounding datagrams.

The same approach followed in this section could be applied to other cases, namely, just identifying the header or metadata indicating the type of data structure, and then applying a simple delta decorrelator (with the adequate symbol size and interleaving values) followed by an entropy coder.

### 4.4. SKM Pre-Processing Stage

The last algorithm we propose is for SKM datagrams. These datagrams do not contain multibeam data as the previous two, although they occupy a relevant percentage of the total size. Hence, it is worth designing a tailored algorithm.

The proposed algorithm is very similar to that of MRZ datagrams, but instead of exploiting the correlation between soundings, we exploit the correlation between sensor measurements, that is, the correlation between each sample in *SensorSamples*. From the format description [13], we know that every sample has $N_B$ bytes, and hence, every $N_B$ bytes, we find the same field from another sample. As the reader may notice, this approach is very similar to the one proposed in the previous section. In this case, as SKM are less relevant than MRZ datagrams, we do not reorder the samples and we opt for a simpler

solution, which gives good enough ratios. Formally, given $N_S$ samples $S_i$ with $N_B$ bytes each:

$$E_i = S_i - S_{i-N_B}, \quad 1 \le i < N_S \tag{14}$$

Note that the first sample $i = 0$ is directly copied to the prediction errors sequence, together with the header and the *InfoPart* structs.

### 4.5. Lossy Compression

Lossy data compression can be selected by the user through the FAPEC command line options or its API parameters, allowing several levels of quality degradation. Lossy compression has only been implemented for MWC amplitude and phase samples. The level of losses can be different in amplitude or phase samples, as decided by the user. In the case of a high-resolution water column phase, samples may be up to 16 bits. For this reason, we provide 16 levels of losses, leading to divisors from 2 to 256. The difference between levels increases exponentially, and hence, we have more granularity for low levels of losses. Furthermore, note that the last level is only intended for high-resolution phase samples, as it eliminates the 8 Least Significant Bits (LSB). It could also be used to completely suppress the low-resolution phase or water column amplitude if the user is not interested in it. This may be useful, for example, in the case of combined KMALL files (with MRZ and MWC datagrams), allowing the generation of compressed files without MWC samples.

Losses are introduced by quantizing each of the original raw samples, that is, dividing the sample value by a given factor, which depends on the quality level given by the user. The resulting value is rounded to the nearest integer. Rounding instead of truncating is significant in terms of quantization error as it reduces its bias. For instance, a white pixel ($+128$) would never be recovered as white if truncation were applied.

This approach translates into a reduction in the number of gray shades of the water column image, from the original 256 shades (8-bit samples) to 128 (level 1) or even just 2 shades (level 15). However, to avoid severe degradation, it is not recommended to use levels above 10 (11 gray shades).

### 4.6. Additional Remarks

Before proceeding to test the proposed algorithm, we shall remark on its robustness in front of unexpected data. The main purpose of the mechanisms here described is to avoid any kind of loss. It is useful in two cases: corrupted KMALL files and non-KMALL files. In practice, both cases are the same, as distinguishing a corrupted datagram from a non-KMALL file would require a continuous checking of data.

The approach we follow makes use of the header information in a KMALL datagram. In the format specification [13], it is established that the first 4 bytes are the datagram size in bytes, the following 4 bytes are the datagram type and the last 4 bytes of the datagram are also the size. This information allows us to perform two checks: first, we validate the datagram type. That is, the first byte must be a hash sign #, and the following three bytes must be capital letters from the Latin alphabet. If this condition is not fulfilled, we can assume that the input is not a KMALL datagram. If this test is successful, we perform a second one, which mainly consists in comparing the sizes given in the header and in the last bytes with the chunk size. If the result is bigger (as an absolute value), we can assume that those fields are not datagram sizes; hence, the input data are non-KMALL. These checks are summarized in Algorithm 1.

If the tests fail and we assume non-KMALL data, we apply a fallback algorithm, which consists in predicting a sample with the previous one for the whole chunk. Note that the advantage of chunking appears again: if a KMALL chunk is corrupted but the next one is correct, then the algorithm proposed in the previous sections will be applied.

---

**Algorithm 1** KMALL datagrams consistency check algorithm.

---

1:  *dgmType ← datagram.header.dgmType*
2:  *dgmBytes ← datagram.header.dgmBytes*
3:  **if** dgmType[0] = # && dgmType[1:4] = [A-Z]+ **then**
4:    **if** |dgmBytes| > chunkSize **then**
5:      *compressFallback()* {Apply a simple differential pre-processing}
6:    **else**
7:      *compressKmall()* {Apply algorithm described in previous sections}
8:    **end if**
9:  **else**
10:    *compressFallback()*
11: **end if**

---

## 5. Test Setup

In the following tests and in the preliminary data analysis, we have used three different datasets: the first one, provided by Kongsberg Maritime, contains 40 files (20 `.kmall` + 20 `.kmwcd`) acquired by an EM304. These files contain 8-bit phase samples. The second one, also provided by Kongsberg Maritime, contains 52 files (26 + 26) acquired by an EM2040 echosounder and six files from an EM712 (3 + 3). The last dataset contains five files from an EM304 acquired by Fugro over a salt dome with known plumes in the Gulf of Mexico. The peculiarity of this last dataset is that both MWC and MRZ datagrams are contained in the same `.kmall` file.

If we want to consider real-time compression, we must take into account the throughput of each MBES and ensure the compression speed (also referred to as compression throughput) is higher. The data rate of an echosounder depends on the ping rate and thus on the water depth (see Section 2). Other features such as the number of beams or the backscatter samples resolution also have an effect. In Table 1, we show the mean data rates and depth in the acquisition of the three datasets.

**Table 1.** Data rates, depth and maximum file sizes for the datasets used in this paper.

| | D1-EM304 | D2-EM712 | D2-EM2040 | D3-EM304 |
|---|---|---|---|---|
| Depth (m) | 190 | 190 | 15 | 1000 |
| KMALL data rate (MB/h) | 492 | 731 | 2448 | 1347 |
| KMALL max. size (MB) | 53 | 123 | 292 | 978 |
| KMWCD data rate (MB/h) | 1932 | 2700 | 3224 | - |
| KMWCD max. size (MB) | 1362 | 453 | 1871 | - |

For the lossless data compression tests, we have decided to compare FAPEC with *gzip*, *bzip2* and *Zstd*, using their default options. *Gzip* is a well-known compressor used in Linux and macOS, equivalent to *Zip*, which uses a combination of LZ77 [18] and Huffman [19] to perform encoding. *Bzip2* is an alternative to *gzip*, often slower but typically leading to better compression ratios. It uses the Burrows–Wheeler transform [20] followed by Huffman coding. Finally, *Zstd* is a compressor, which typically yields ratios comparable to those of *gzip*, but much faster. It combines LZ77, Finite State Entropy [21] and Huffman coding.

Note that, in the market, there are no lossy algorithms for KMALL data except for the one we present in this paper. For this reason, lossy compression has been evaluated by means of ratios, quality (PSNR) and visualization.

For a fair comparison of compressing time, we have forced a single thread operation for FAPEC and *Zstd*. *Gzip* and *bzip2* already operate with one thread by default. All tests were run on a computer with Intel Xeon CPU E5-2630 v3 2.40 GHz running 64-bit CentOS 7.9. The complexity of the algorithm has been evaluated using Valgrind.

From all the files in our dataset, we have chosen two (one from the EM2040 dataset and one with phase from the EM304 dataset), and we have plotted the histogram of the

prediction errors after the pre-processing step (before the entropy coder). It allows us to evaluate the performance of the algorithm without the constraints of any entropy coder.

## 6. Test Results

### 6.1. Algorithm Complexity

In this section, we compare the complexity of the algorithm for the WCD format (see [12]) and the one here proposed. In order to do so, we compute the number of instructions per byte (I/B) for both algorithms. The result is that the former algorithm needs, on average, 74 I/B, whereas the latter only needs 46 I/B (see Figure 8). The main reason is that the old format required a very tailored algorithm, but thanks to the optimized definition of the KMALL format, processing is now much easier. This comparison is fair because we only take into account water column amplitude samples, which are quantized as `int_8` in both formats. In this comparison, the percentage of MWC datagrams is over 99% in all cases; hence, using the file ratio is unbiased. For the same reason, the combined files from Fugro have not been included.

It is also worth pointing out the correlation between the compression ratio and the number of instructions per byte. In general, we expect that files with a lower ratio need more instructions, as more bytes need to be written. In addition, for flat distributions, the FAPEC core needs to check more conditions for the different entropy levels. In Figure 8, we show this trend both in the algorithm from [12] and the one we propose. Note that the number of instructions per byte in the best case of [12] is still higher than the number of instructions in the worst case of our implementation. Although it seems that WCD has, in general, better ratios than KMWCD, the behavior we observe is scene-specific, as we will see in the following section when comparing FAPEC to other compressors.

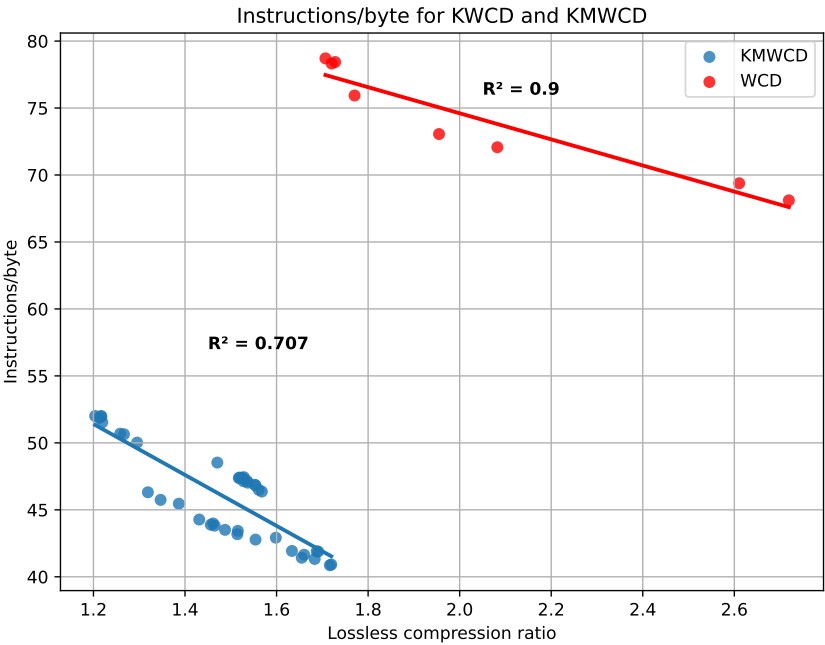

**Figure 8.** Relationship between ratios and instructions per byte for the KWCD and the KMWCD algorithms. Fugro files not included.

Note that in this section, we have just compared the complexity of the water column algorithms. For the sake of completeness, we have also evaluated the MRZ algorithm (see Section 4.3), and it only needs 35 I/B, on average.

### 6.2. Lossless Compression

In this section, we show the compression ratio and compression throughput obtained in the tests for KMWCD and KMALL files. The compression ratio is defined as the original

file size divided by the compressed file size and the compression throughput as the raw data volume compressed per second. The lossless compression results are presented in three sections: the first one for `.kmwcd` files (that is, MWC datagrams), the second one for `.kmall` files (that is, MRZ and SKM datagrams) and the third one for combined files (that is, MWC, MRZ and SKM datagrams). Regarding compression speed, it determines if a compressor is capable of performing real-time compression. In practice, real-time compression may be performed by an Autonomous Underwater Vehicle (AUV) with an ARM processor with much less computing power than our testing setup.

### 6.2.1. MWC Datagrams

As can be seen in Figure 9, FAPEC achieves the best compression ratios, except for two of the eleven scenes from EM2040. The improvement is especially notorious in EM712 and EM304 files. It is worth mentioning that higher ping rates, in general, lead to better compression ratios. This is an expected result, as a high sampling rate means a higher correlation between samples, which is exactly what FAPEC exploits.

When compared to *gzip*, FAPEC yields ratios at least 12% better, excluding files with phase samples, reaching even 33% for some EM712 files. On average, FAPEC performs 22% better than *gzip* and *Zstd*, which share very similar compression ratios. On the other hand, *bzip2* ratios for EM2040 files are close to those of FAPEC, and even slightly better in two situations. In this case, FAPEC is 9% better on average. One could consider using *bzip2*, but in addition to the compression ratio, we should take into account the speed results. Furthermore, FAPEC results are outstanding. It is three times faster than *gzip* and six times faster than *bzip2*. Regarding *Zstd*, it is twice as fast as FAPEC, but as we saw earlier, the compression ratios are rather poor. We should remark that even with the most aggressive compression levels, FAPEC yields better ratios and *Zstd* becomes slower than *bzip2*.

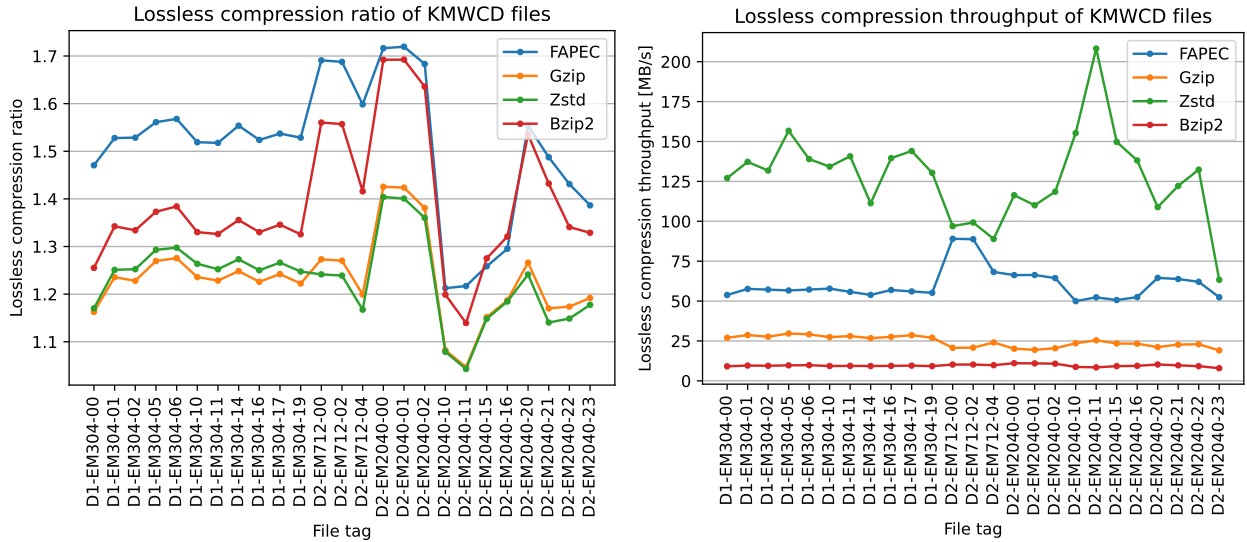

**Figure 9.** Lossless compression ratio (**left** panel) and lossless compression throughput (**right** panel) for the tested KMWCD files.

Clearly, FAPEC is the most optimized option to compress water column data. This is also observed in Figure 10, where we show a scatter plot of the process time (in seconds) and the inverse compression ratio for each file and compressor. Note that, in this case, the compression ratio has been defined as the compressed size divided by the original size; hence, the closer to the origin, the better. In order to facilitate visualization, the time has been cut at 65 s, which implies that some *gzip* and *bzip2* realizations are not shown in the figure.

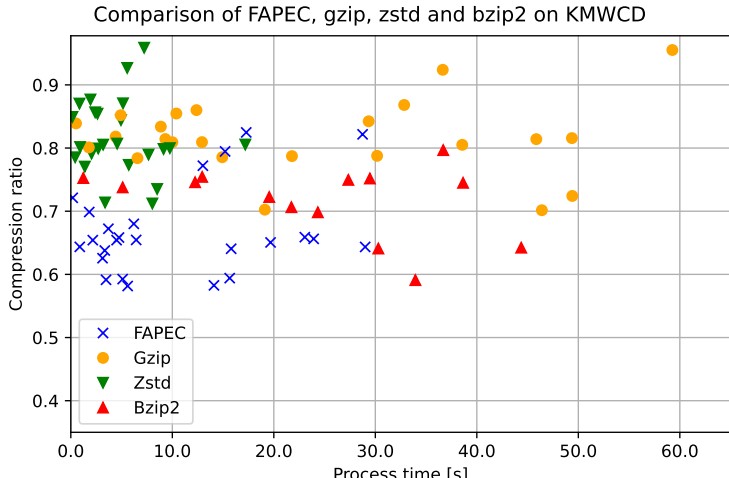

**Figure 10.** Comparison of time and ratio performance of lossless compressors for KMWCD files.

### 6.2.2. MRZ and SKM Datagrams

The performance of FAPEC in KMALL files is even better than that in KMWCD files. It is at least 28% better than *bzip2* and 35% on average. In this case, *bzip2* does not outperform *gzip* and *Zstd*, and all three exhibit similar performance in terms of ratio.

Similar to the previous section, we have also plotted the throughput in Figure 11. The simplicity of our algorithm allows FAPEC to be even slightly faster than *Zstd*. Compared to *gzip* or *bzip2*, FAPEC is between 5 and 6 times faster than the former and between 9 and 10 times than the latter. Finally, the outperformance of FAPEC is also observed in the time and inverse compression ratio scatter plot in Figure 12. In this case, the time axis has been cut at 10 s.

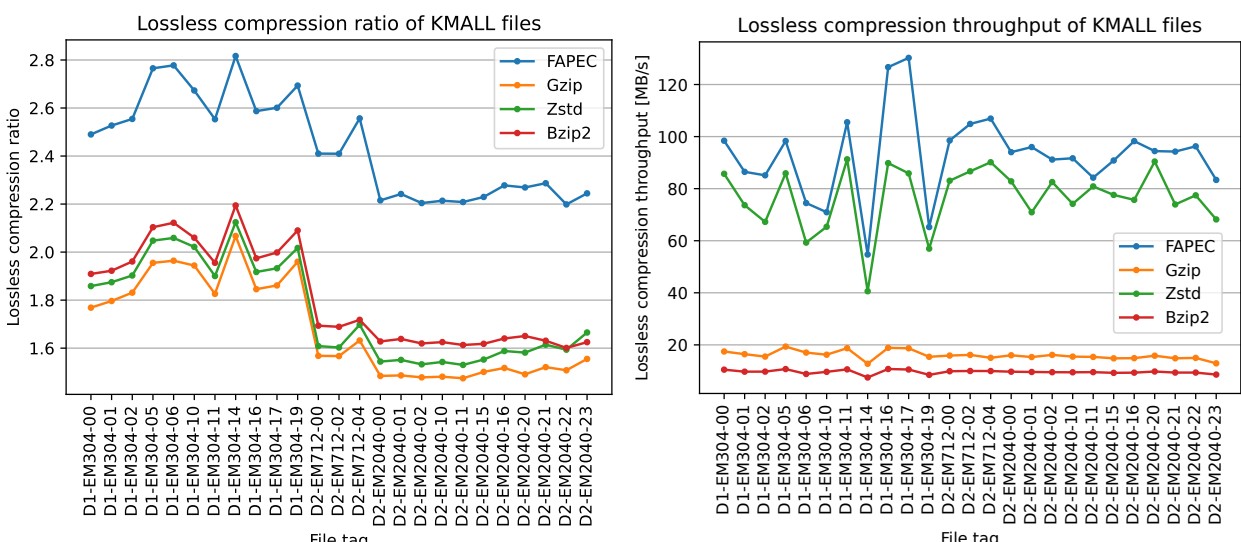

**Figure 11.** Lossless compression ratio (**left** panel) and lossless compression throughput (**right** panel) for the tested KMALL files.

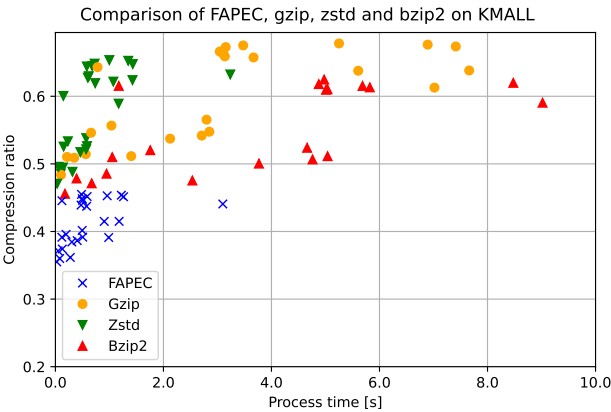

**Figure 12.** Comparison of lossless compressors showing that FAPEC performs better both in time and ratio than *gzip*, *Zstd* and *bzip2* on KMALL files.

### 6.2.3. Combined Files

The dataset provided by Fugro has been excluded from the previous two sections, as it contains both MRZ (and SKM) and MWC datagrams. In the five files, the datagrams distribution is approximately: 70% MWC datagrams, 20% MRZ datagrams and 10% SKM datagrams.

Once again, FAPEC achieves the highest compression ratios. In this case, it is even more significant, as it shows at least 50% better performance than *bzip2*, the second-best option. *Zstd* and *gzip* exhibit a very similar performance, and in this case, also similar to *bzip2*.

Regarding the throughput, FAPEC is three times faster than *gzip* and seven times faster than *bzip2*. However, *Zstd* is slightly faster than FAPEC, as can be seen in Figure 13.

Note that this dataset is especially interesting because high ratios are obtained in the presence of mostly MWC datagrams. These results strengthen our observation from Section 6.1, where we remarked that the low ratios in Figure 8 are scene-specific.

It is worth mentioning that we also carried out these tests on a Raspberry Pi 400 platform, featuring an ARM processor running at 1.8 GHz, to evaluate the compression speeds of the various solutions. Here, FAPEC is undoubtedly the fastest data compressor, either in these combined files or in the separate ones, reaching compression throughputs of 35 to 75 MB/s, whereas the other compressors can barely reach 10 MB/s.

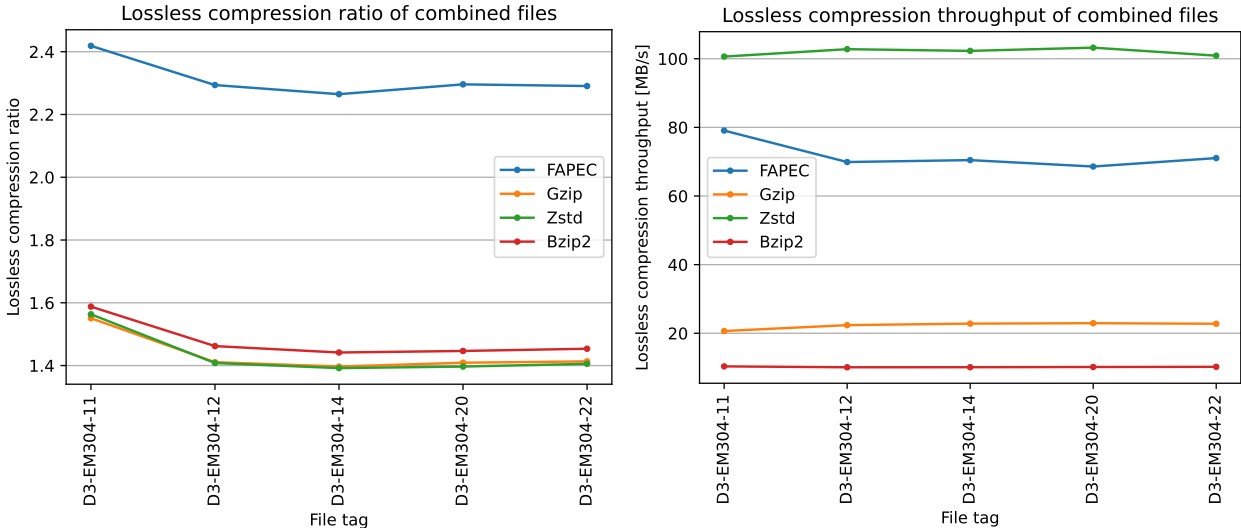

**Figure 13.** Lossless compression ratio (**left** panel) and lossless compression throughput (**right** panel) for the tested KMALL combined files.

### 6.3. Lossy Compression

The lossy algorithm we propose is often called *near lossless*, as the quantization error is bounded. In this section, we evaluate the quality of lossy compressed files, both visually and by means of the PSNR, and we show the ratio increase between levels.

When evaluating lossy compression algorithms, it is recommended to calculate the Peak Signal to Noise Ratio (PSNR), a metric to quantify the reconstruction quality for files subject to lossy compression. It is usually defined via the Mean Squared Error (MSE):

$$PSNR = 10 \cdot \log_{10}\left(\frac{MAX_I^2}{MSE}\right),$$ (15)

where $MAX_I^2$ is the maximum value of the sample.

The PSNR is usually evaluated in decibels (dB), with typical values between 30 and 50 dB (where higher is better) for 8-bit samples, as in our case. Figure 14 shows the relationship between different PSNR results at a given lossy level. It is worth mentioning that the near lossless approach followed implies that the PSNR values obtained with FAPEC for any file are grouped around some specific values. Note that this statement is only true if the files contain the same data types. That is, files with phase samples and files without phase samples are not comparable (see Figure 14) between them, as both the applied algorithm and the data structures differ.

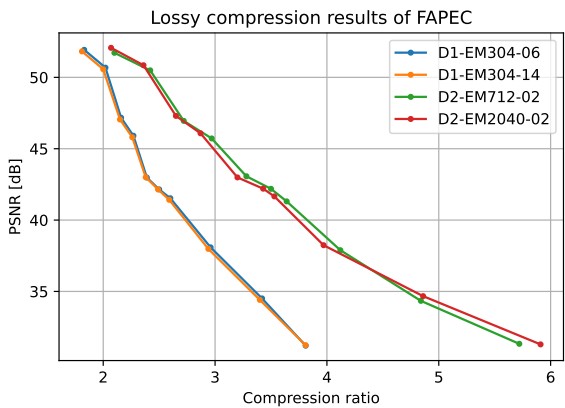

**Figure 14.** Quality (PSNR) of the reconstructed water column files at different FAPEC lossy levels.

In addition to the quantitative results, Figure 15 provides a qualitative evaluation of the lossy compression. It corresponds to an EM304 survey in the Gulf of Mexico. The lossy levels of FAPEC illustrate the effect of the quantization, leading to images with lower contrast. It is worth mentioning that, even with a very high level of losses (level 10, eleven shades), we can still clearly identify the gas flare. For moderate levels of losses, such as level 4 (52 shades), the degradation is barely noticeable, but the compression ratio increases from 2.27 (lossless) to 3.43 (level 4). Note that level 16 would completely suppress the water column data, allowing us to only store the bathymetry information in the combined files.

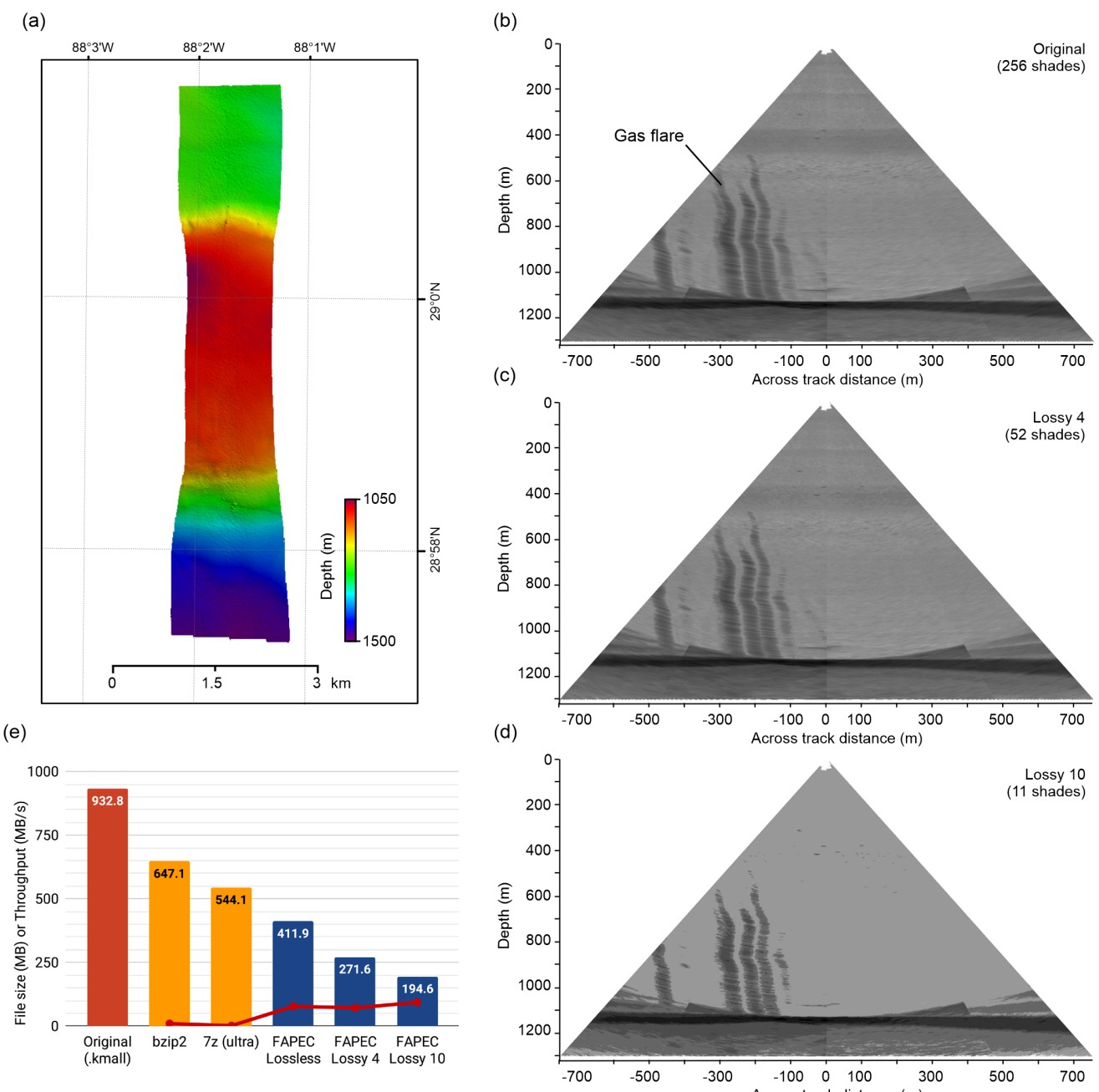

**Figure 15.** Kongsberg EM304 survey in the Gulf of Mexico. (**a**) Bathymetry. (**b–d**) Mid-water backscatter showing gas flares at different levels of lossy compression. (**e**) Original and compressed file sizes. The red line and dots indicate the compression rate. Data courtesy of Fugro USA Marine.

Figure 15e shows the compressed file sizes obtained with FAPEC, using its lossless option as well as the two lossy levels evaluated here. For comparison, we also include the raw `.kmall` size, as well as the compressed size when using *bzip2*. For further completeness, in this case, we also show the size achieved by the *7z* compressor in its "ultra" configuration, which is even slower than *bzip2*. As can be seen, not even this option can approach FAPEC. The red line and dots indicate the compression throughput (in MB/s) for the different options, which confirms that FAPEC is much faster.

### 6.4. Pre-Processing Stage Performance

This section is focused on evaluating the goodness of compressible data after the decorrelation stage. Our first approach is to plot a histogram of the original samples together with a histogram of the prediction errors (i.e., the values that will be sent to the FAPEC entropy coder). For a better visualization, the bin width is calculated from the number of samples, and it is the same for both histograms. In addition, the prediction errors histogram also includes three vertical lines at percentiles 5, 50 and 95 in order to clearly show the range where 90% of the data is contained.

Making use of the histograms, the cumulative histograms of the original samples and the prediction errors are calculated and plotted together. Although these graphics are just the integral of the histograms described above, they provide a clear way to observe sample concentrations in intervals.

Figures 16 and 17 show the histograms and cumulative histograms of one file from the EM2040 dataset (D2-EM2040-22) and one file with 8-bit phase samples from the EM304 dataset (D1-EM304-11).

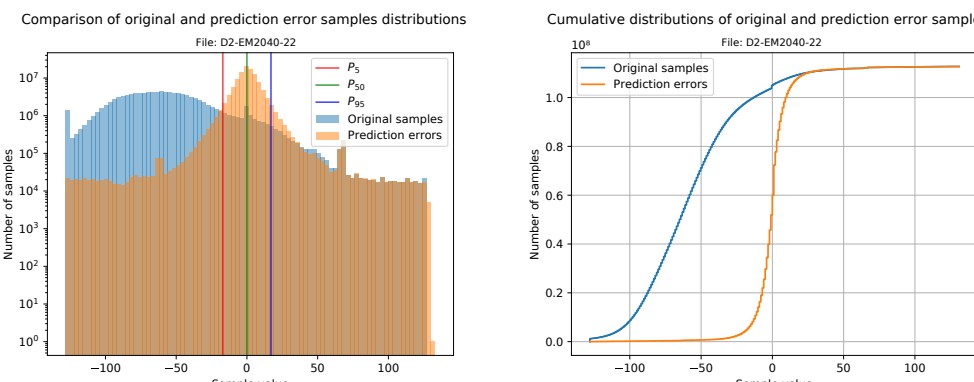

**Figure 16.** Histograms showing the range reduction in water column data from an EM2040.

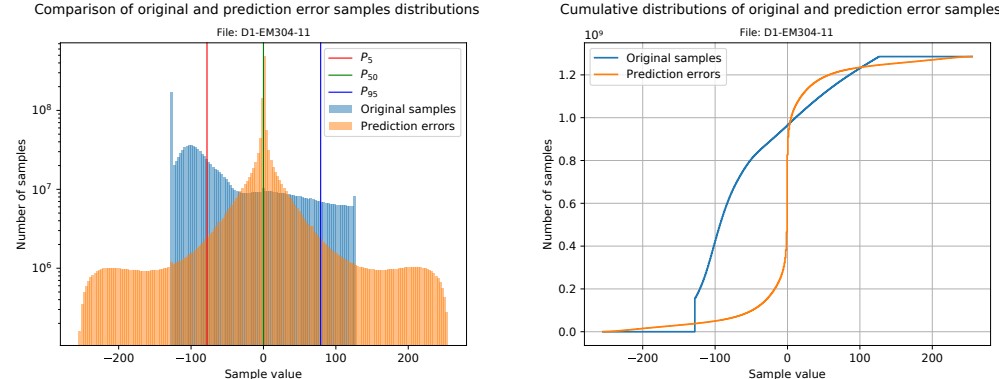

**Figure 17.** Histograms showing the range reduction in water column data from an EM304.

From the histograms, we see that original data are distributed among the whole dynamic range, which would result in a poor compression ratio. After the pre-processing stage, the dynamic range is heavily reduced. Clearly, Figure 16 shows better results than Figure 17. The reason is that D1-EM304 contains 8-bit phase samples, and as we mentioned in Section 4.2, they are too difficult to process.

### 6.5. Decompression Performance

As shown in the previous sections, FAPEC, with its new KMALL pre-processing stage, provides an excellent compression speed, only comparable to *Zstd*. This is where we have allocated more effort, aiming at its applicability to remote sensing devices such as

autonomous vehicles running on batteries, or in general, with limited computing power. However, decompression speed should also be high, allowing to directly process or visualize FAPEC-compressed files on the fly, for example.

We have used the same testing environment to evaluate the decompression speed of the different data compression solutions considered here. In general, *Zstd* is clearly the fastest single-thread decompressor, reaching speeds around 500 MB/s on this platform. Both FAPEC and *gzip* exhibit similar decompression speeds on `.kmwcd` and combined `.kmall` files (around 100 MB/s), although the FAPEC decompression speed almost doubles on non-combined `.kmall` files (that is, without watercolumn datagrams). Finally, *bzip2* is clearly the slowest option, also in decompression, barely reaching 20 MB/s.

## 7. Conclusions and Future Work

In this paper, we have proposed a data compressor pre-processing stage for bathymetry and water column data acquired by Multibeam Echosounders (MBES), adapted to the KMALL format from Kongsberg Maritime, but easily adaptable to other vendors as well. From all the systems that acquire information from the seafloor and water column, MBES are probably the most demanding in terms of processing power and data acquisition rates, being able to acquire several gigabytes of information per hour. The use of efficient data compression algorithms allows to drastically reduce the storage needs and to speed up and lighten data transfer in remote operations that require relatively slow and expensive narrowband satellite transmission, hence reducing the associated costs.

We have tested the bespoke stage with the FAPEC entropy coder on three datasets formed by different echosounder models: Kongsberg Maritime EM304, EM712 and EM2040. When comparing FAPEC with the general-purpose compressors *gzip*, *Zstd* and *bzip2*, we obtain, on average, the best compression ratios on water column data. Additionally, FAPEC always obtains remarkably better ratios in bathymetry data. This is even more notorious in combined bathymetry and water column files such as the ones in the third dataset (D3-304). Regarding compression time, *Zstd* is faster when compressing water column data, although the compression ratios are clearly worse. For bathymetry data, FAPEC is even faster than *Zstd*. Lossy compression has also been treated, but only for water column datagrams. The level of losses is selectable by the user, with each level reducing the number of gray shades, hence the contrast. We have shown that even at high levels with only 11 shades, important features such as gas flares are still detectable. This option provides a powerful tool for water column exploratory surveys. FAPEC is a valuable asset, particularly for remote technical assistance and monitoring of crewed vessels and autonomous vehicles.

During the study presented in this paper, we have identified some possible improvements that could increase the compression ratios, yet some of them would also increase the compression time. For instance, the algorithm for SKM could be improved in the sense that the prediction errors for common fields are stored in a buffer and sent together to the entropy coder. This approach is the one followed for MRZ datagrams, which are more relevant in terms of percentage. On the other hand, there are several datagram types that are not pre-processed. For these, instead of sending them directly to the entropy coder, a generic compression strategy could be applied, for instance, the Lempel-Ziv-Welch (LZW), which is already implemented as a FAPEC stage.

**Author Contributions:** Conceptualization, J.P., A.M. and D.A.; methodology, J.P.; software, A.M. and J.P.; validation, J.R., D.A. and J.P.; formal analysis, A.M.; investigation, A.M. and J.P.; resources, J.P.; data curation, J.P.; writing—original draft preparation, A.M.; writing—review and editing, M.V., F.d.C., D.A., J.P. and G.M.; visualization, A.M., D.A., M.V. and F.d.C.; supervision, J.P. and J.R.; project administration, J.P. and D.A.; funding acquisition, J.P. and D.A. All authors have read and agreed to the published version of the manuscript.

**Funding:** This work was (partially) funded by the ERDF (a way of making Europe) by the European Union through grant RTI2018-095076-B-C21, the Institute of Cosmos Sciences University of Barcelona (ICCUB, Unidad de Excelencia María de Maeztu) through grant CEX2019-000918-M, the Spanish State Research Agency (PID2020-114322RBI00), the European Union's Horizon 2020 research and

innovation programme (Marie Sklodowska-Curie grant 658358), the Catalan Government Excellence Research Groups Grant to GRC Geociències Marines (ref. 2017 SGR 315), the Spanish Ministry of Science and Innovation project RODIN (PID2019-105717RB-C22/AEI/10.13039/501100011033), and Fellowship FI 2019 by the Secretary for University and Research of the Generalitat de Catalunya and the European Social Fund.

**Data Availability Statement:** Restrictions apply to the availability of these data. Data were kindly provided by Kongsberg Maritime and Fugro USA Marine Inc. and are available from the authors with the permission of Kongsberg Maritime or Fugro USA Marine Inc.

**Acknowledgments:** The authors would like to thank Terje Haga Pedersen (Kongsberg Maritime) for providing some of the datasets used in this paper.

**Conflicts of Interest:** The authors declare the following conflict of interest. J.P. is CTO of DAPCOM, original developer of the FAPEC algorithm. A.M. is a research engineer at DAPCOM. The funders had no role in the design of the study; in the collection, analyses, or interpretation of data; in the writing of the manuscript; or in the decision to publish the results.

## Abbreviations

The following abbreviations are used in this manuscript:

| | |
|---|---|
| FAPEC | Fully Adaptive Prediction Error Coder |
| MBES | Multibeam Echosounders |
| WCD | Water Column Data |
| UDP | User Datagram Protocol |
| MWC | Multibeam Water Column |
| MRZ | Multibeam Raw Range and Depth |
| SKM | Sensor Attitude Data |
| LSB | Least Significant Bits |
| IQ | In-phase and Quadrature |
| PSNR | Peak Signal to Noise Ratio |
| MSE | Mean Squared Error |
| LZW | Lempel-Ziv-Welch |
| AUV | Autonomous Underwater Vehicle |
| API | Application Programming Interface |

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
