# Peer review of "Compression of Multibeam Echosounders Bathymetry and Water Column Data"

_remotesensing, doi:10.3390/rs14092063_

Round 1

Reviewer 1 Report

Interesting article to overcome the problem of transmission and storage of large files.

In my opinion some figures could be improved.

Fig2 and  Fig. 7could be made clearer perhaps with some arrows and labels. text is clear but the image not much

In Fig.5 the rapresentation of datagram is not clear because does not correspond much to the text. the second part is InfoPart and the third Sensor Samples. in the text " sensor attitude data" is wrote

In cap 4.5 lossy compression, (line 306) it's not clear to me where lossy data compression can be selected by the user.

Reviewer 2 Report

This manuscript submitted to Remote Sensing- Compression of Multibeam Echosounders Bathymetry and Water Column Data- provided a very interesting read. The value of this proposed technique for handling large file sizes obtained from recording the full water column backscatter is a very relevant topic in this area of acoustics research at present.

My expertise in compression algorithms is very limited so I am unable to comment on the technical aspects of the algorithms, but the paper is well presented, and I could follow the approach. I can see the value of this method for transferring data to shore. I can also see how the lossy approach might be interesting for downsampling data for easier post-processing but am not so sure how the lossless approach would be used other than for data transfer, unless there is software that works better with the compressed files (or perhaps I'm missing something here that is key to understanding this approach?).

In terms of improving the paper I advise that some terminology could be better defined. Particularly the use of the word "scene" (first on line 204). I assume the authors mean an image of a single ping, but they use the term "scene-specific" a few times and this is ambiguous to me (perhaps jargon from a different field?).

Figures 2 and 7 should have axes to make them more clear, I assume it is samples/time on X and beams on Y, but this is not obvious to the reader (probably especially for someone that doesn't spend their life looking at WCD images! ).  It would also help if they were labelled in relation to their algorithm terminology (e.g., beam=j, samples=i). It took me awhile to orient myself to what the authors were doing and would be much easier if this were clear in the figures.

To try to understand better what the authors have done for the WCD/MWC algorithm (section 4.2) I looked up their 2019 paper (ref 12) to see how they originally made the algorithm. It looks like in that variation they handle the data differently below the MSR (or "parabola" as they call it). So, the "rectangle" above the MSR gets the prediction error calculated based on the data at the same sample range at the neighbouring beam, but for data below MSR (including bottom detections and sidelobes, if I understand correctly) gets calculated with a slightly different empirically derived formula. In section 4.2 of the current manuscript, the authors show the 2019 formulas (4-7) and then propose an improvement in equations 8-9 which applies the same algorithm to samples both above and below MSR.

[Side note: to be very picky they switch between calling the above MSR area "rectangular" and "square"- that should be consistent, I assume square is the same as rectangle which refers to above MSR].

 Although in Section 6.1 they describe a comparison between the 2019 algorithm and the current one, this is based on "complexity" of the algorithm. I am curious as to how the results of the current algorithm that treats all data above and below MSR with mostly a single equation compares to their previous results- perhaps this is something they could cover either in this section or in the discussion? I assume being a lossless approach the end result (decompressing) should be exactly the same, but it would be interesting to know a bit more of how that improved their approach and what the impact of this improvement had.

One additional thing is that I would be interested to know the size of the files that the authors are compressing, especially the ones they are using to compare different compressing speeds. I think that this should be added to the discussion.

Overall, this is a nicely written paper and looks like it provides a very promising method of compressing KMALL files. The manuscript would be improved by adding these elements listed above to the discussion and by also adding a short explanation about how this method might be applied and used in operational activities (i.e in practice).
